# Bringing Onco-Innovation to Europe’s Healthcare Systems: The Potential of Biomarker Testing, Real World Evidence, Tumour Agnostic Therapies to Empower Personalised Medicine

**DOI:** 10.3390/cancers13030583

**Published:** 2021-02-02

**Authors:** Denis Horgan, Gennaro Ciliberto, Pierfranco Conte, Giuseppe Curigliano, Luis Seijo, Luis M. Montuenga, Marina Garassino, Frederique Penault-Llorca, Fabrizia Galli, Isabelle Ray-Coquard, Denis Querleu, Peter Riegman, Keith Kerr, Hein Van Poppel, Anders Bjartell, Giovanni Codacci-Pisanelli, Jasmina Koeva-Balabanova, Angelo Paradiso, Zorana Maravic, Vassiliki Fotaki, Nuria Malats, Chiara Bernini, Simonetta Buglioni, Alastair Kent, Elisabetta Munzone, Ivica Belina, Jan Van Meerbeeck, Michael Duffy, Beata Jagielska, Ettore Capoluongo

**Affiliations:** 1European Alliance for Personalized Medicine, Avenue de l’Armee/ Legerlaan 10, 1040 Brussels, Belgium; Chiara.Bernini@euapm.eu; 2IRCCS Istituto Nazionale Tumori “Regina Elena”, Via Elio Chianesi, 53, 00128 Rome, Italy; gennaro.ciliberto@ifo.gov.it (G.C.); simonetta.buglioni@ifo.gov.it (S.B.); 3Dipartimento di Scienze Chirurgiche Oncologiche e Gastroenterologiche, University of Padova, Via Giustiniani 2, 35128 Padova, Italy; pierfranco.conte@unipd.it; 4Department of Oncology and Hemato-Oncology, University of Milano and European Institute of Oncology, IRCCS, 20139 Milano, Italy; giuseppe.curigliano@ieo.it; 5Pulmonary Department, Clínica Universidad de Navarra, Calle Marquesado de Sta. Marta, 1, 28027 Madrid, Spain; lseijo@unav.es; 6Ciber Enfermedades Respiratorias (CIBERES), Av. de Monforte de Lemos, 3-5, 28029 Madrid, Spain; 7Center for Applied Medical Research (CIMA), Schools of Sciences and Medicine, University of Navarra, Av. de Pío XII, 55, 31008 Pamplona, Spain; lmontuenga@unav.es; 8CIBERONC, Av. Monforte de Lemos, 3-5, 28029 Madrid, Spain; nmalats@cnio.es; 9S.S. Oncologia Medica Toraco Polmonare, Fondazione IRCCS Istituto Nazionale dei Tumori Via Giacomo Venezian, 1, 20133 Milano, Italy; marina.garassino@istitutotumori.mi.it; 10Department of Pathology and Tumor Biology, University of Clermont-Auvergne, 49 bd François Mitterrand, 63001 Clermont-Ferrand, France; frederique.penault-llorca@clermont.unicancer.fr; 11Associazione aBRCAdaBRA Onlus Via Volontari Italiani del Sangue, 32, 90128 Palermo, Italy; fabrizia.galli@materdomini.it; 12Medical Oncology Department, Centre Leon Bérard & Université Claude Bernard Lyon, 69008 Lyon, France; isabelle.ray-coquard@lyon.unicancer.fr; 13Surgery Department, Institut Bergonié Cancer Center, Centre Léon Bérard Cheney D- 2 ème étage -28 Rue Laennec, 69373 Lyon, France; denis.querleu@esgo.org; 14Department of Pathology, Josephine Nefkens Institute, Erasmus Medical Center, Be 235b, Dr Molwaterplein 50, 3015 Rotterdam, The Netherlands; p.riegman@erasmusmc.nl; 15Department of Pathology, University of Aberdeen, King’s College, Aberdeen AB24 3FX, UK; Keith.Kerr@quinnipiac.edu; 16Department of Urology, University Hospitals Leuven, Herestraat 49, 3000 Leuven, Belgium; Hendrik.vanpoppel@uzleuven.be; 17Department of Urology, Skane University Hospital, Box 117, 221 00 Lund, Sweden; anders.bjartell@med.lu.se; 18Department of Medical and Surgical Sciences and Biotechnology, University of Rome, “la Sapienza”, Piazzale Aldo Moro, 5, 00185 Roma, Italy; Giovanni.Codacci-Pisanelli@uniroma1.it; 19Bulgarian Association for Personalised Medicine, 45 Bacho Kiro Str., 1202 Sofia, Bulgaria; office.bappm@gmail.com; 20Scientific Directorate, IRCCS Istituto Tumori Giovanni Paolo II, Viale Orazio Flacco, 65, 70124 Bari, Italy; a.paradiso@oncologico.bari.it; 21Digestive Cancers Europe, Rue de la Loi 235, 1040 Brussels, Belgium; zorana.maravic@gmail.com (Z.M.); vfotaki@ed.ac.uk (V.F.); 22Spanish National Cancer Research Centre (CNIO), Calle de Melchor Fernández Almagro, 3, 28029 Madrid, Spain; 23Independent Patient Advocate, 14 Farthing Road Downham Market, Norfolk PE38 0AF, UK; info@alastairkent.com; 24Division of Medical Senology, IEO, European Institute of Oncology IRCCS, Via Ripamonti 435, 20141 Milano, Italy; elisabetta.munzone@ieo.it; 25KUZ-Coalition of Association in Healthcare, Trpimirova 11, 10000 Zagreb, Croatia; ivica.belina50@gmail.com; 26Thoracic Oncology-MOCA, University Hospital Antwerp, Wilrijkstraat 10, 2650 Edegem, Belgium; jan.van.meerbeeck@uza.be; 27UCD School of Medicine, Conway Institute of Biomolecular and Biomedical Research, University College Dublin, Belfield, Dublin 4, Ireland; michael.j.duffy@ucd.ie; 28Maria Skłodowska-Curie Institute of Oncology, Wawelska 15 B, 00-001 Warszawa, Poland; beata.jagielska@pib-nio.pl; 29Department of Molecular Medicine and Medical Biotechnologies, University of Naples Federico II, 80131 Naples, Italy; 30CEINGE-Biotecnologie Avanzate, Via Gaetano Salvatore, 486, 80131 Napoli, Italy

**Keywords:** Europe’s Healthcare Systems, biomarkers, oncogenomics, HRD, cancers

## Abstract

**Simple Summary:**

The increasing number of data supporting use of a personalized approach in cancer treatment, is changing the path of patient’s management. In the same time, the availability of technologies should allow patients to receive the best test for the specific individual condition. This is theoretically true, when a specific test is designed for the specific disease condition, while it is difficult to implement in the setting of agnostic therapies. Financial sources availability related to the non homogeneous health systems working in the different countries do not allow for an immediate implementation of the technologies and test commercially available. Future perspectives for targeted oncology include tumor-agnostic drugs, which target a given mutation and could be used in treating cancers from multiple organ types. Therefore, the present paper is aimed to both underline a how much important is this new view and also to sensitize the international bodies that supervise health policies at the decision-making level, with the aim of harmonizing cancer treatment pathways in at least all European countries.

**Abstract:**

Rapid and continuing advances in biomarker testing are not being matched by uptake in health systems, and this is hampering both patient care and innovation. It also risks costing health systems the opportunity to make their services more efficient and, over time, more economical. The potential that genomics has brought to biomarker testing in diagnosis, prediction and research is being realised, pre-eminently in many cancers, but also in an ever-wider range of conditions—notably *BRCA1/2* testing in ovarian, breast, pancreatic and prostate cancers. Nevertheless, the implementation of genetic testing in clinical routine setting is still challenging. Development is impeded by country-related heterogeneity, data deficiencies, and lack of policy alignment on standards, approval—and the role of real-world evidence in the process—and reimbursement. The acute nature of the problem is compellingly illustrated by the particular challenges facing the development and use of tumour agnostic therapies, where the gaps in preparedness for taking advantage of this innovative approach to cancer therapy are sharply exposed. Europe should already have in place a guarantee of universal access to a minimum suite of biomarker tests and should be planning for an optimum testing scenario with a wider range of biomarker tests integrated into a more sophisticated health system articulated around personalised medicine. Improving healthcare and winning advantages for Europe’s industrial competitiveness and innovation require an appropriate policy framework—starting with an update to outdated recommendations. We show herein the main issues and proposals that emerged during the previous advisory boards organised by the European Alliance for Personalized Medicine which mainly focus on possible scenarios of harmonisation of both oncogenetic testing and management of cancer patients.

## 1. Introduction: Healthcare Efficiencies

Europe is missing out on major opportunities. The rapid growth in the availability of biomarker testing based on molecular diagnostics has made it possible to improve European citizens’ health. Widespread adoption would support continued improvements in outcomes for diseases, already for cancer, and increasingly across a wide range of conditions. And the prospects are even more attractive if the development and deployment of more sophisticated biomarkers is encouraged. This would enable exploiting the possibilities of precision medicine—and would drive the evolution of much-needed debate and definition of the role of real-world evidence. The impact on patients’ health would be more informed treatment decisions and increased access to targeted treatments, with improved outcomes. The impact on health systems would be the better deployment of resources that precision medicine allows [1]. We report herein the proposals that emerged during the previous advisory boards on this topic. In this regard, some European experts were invited to join the meetings organized by the European Alliance for Personalized Medicine in order to share the main criticisms regarding the oncogenetic testing and possible actions to solve them. None of the participants had any conflicts of interest because the only objective of the present teamwork is to achieve the harmonization of clinical-diagnostic path of overall cancer patients.

## 2. Minimum and Optimum Testing

The short-term need is for an “essential package” of high-quality and timely biomarker testing across the EU for diagnosis and prevention for all cancer patients. This would be backed by the certainty of an adequate reimbursement, and up-to-date guidance to clinicians who had consistent access to clinically meaningful testing results based on long-term validated biomarkers such as EGFR, HER2, and ALK [2].

Further ahead, an optimum scenario should see multi-modality clinical teams, adequately trained in molecular diagnostics, delivering a high standard of care all over Europe, and increasingly for patients with other conditions. Pan-cancer research would operate on the basis of standardized data between multiple platforms, and registries with opt-in and-out criteria for each disease area would provide more robust data sets—including the patient’s disease history, outcomes, treatments and genomic aspects, and information on the use of biomarker tests themselves, such as how often and on which patients [3]. Fuller use of real-world evidence would provide support for acceptance, and prospects would mature for wider endorsement of tumour agnostic therapies where genomic data would allow a focus irrespective of the tumour site of origin [4].

In both paradigms, the aim is to secure the best standard of care for patients. Even the minimum scenario presupposes readiness to overcome the objective national and local limitations of resources, human and financial capital and infrastructure. The longer-range question is how willingly—and how soon—the EU will deploy adequate resources and political will to embrace an optimum biomarker scenario.

## 3. Gaps, and Need for Action

At present, progress is impeded by a wide range of barriers.

Although the scientific community has risen well to the innate scientific challenges of developing sophisticated biomarkers, operational barriers persist in outdated regulations, inadequate infrastructure for data collection and laboratory analysis, insufficient training of healthcare professionals, and fragmented approval and funding systems. Technologies, disease areas, and patient populations vary across Europe, and the apparatus for product approval of biomarkers is under-developed. Inconsistency persists among evidence frameworks for diagnostics, and among standards for demonstrating clinical utility, prolonging uncertainty on reimbursement arrangements.

Health systems are unfamiliar with complex testing systems. Reimbursement of biomarker tests that are matched to treatments with positive clinical trial data, or for patients with single mutations, presents fewer challenges because of the link to a single medicine. Nevertheless, agreement is more elusive when assessing value in molecular diagnostics monitoring disease or patients’ health, or that prognosticate future disease progression or that predict whether a patient will have an adverse reaction to a medicine. And reimbursement remains still more of a challenge, partly because of variations among key stakeholders in background knowledge and literacy on biomarker testing—and in motivation to learn. A general requirement in most countries and regions for new tests to save costs complicates progress still further [5,6,7].

The uncertainty on access, funding and uptake of biomarkers based on molecular diagnostics has a direct negative influence on investment decisions, and impedes innovation and its integration. The disproportion of attention is striking: Diagnostics account for less than 2% of total healthcare spending—but they influence 60% of clinical decision making [8]. Current policy in Europe -and particularly in respect of reimbursement—fails to take this into account, and urgently needs updating [9,10].

## 4. Upcoming EU Policy Changes

The coronavirus pandemic has given new prominence and impetus to health in EU policy, but already there was growing consciousness of the combined perils of sub-optimal health services in the face of demographic change and chronic disease, and of the EU losing out at world level to the scientific innovation on which its prosperity largely depends [6].

This has given rise to EU plans for a Cancer Mission [7], a Beating Cancer plan, a European Health Data Space [9,10] that can exploit volumes of data from real world evidence through exchange among national health data organisations, and the promise of a new EU pharmaceutical strategy by the end of 2020. These initiatives—now complemented by the urgent and heavily-funded search for vaccines and treatments to counter Covid-19—will depend heavily for their success on the extent to which sensitive and sophisticated testing is available—in other words, for greater use of biomarkers.

## 5. Need for Updating 2003 Recommendations

Nearly two decades have passed since the emergence in 2003 of EU recommendations relating to early diagnosis and screening, and in that time the possibilities for better and more interventive healthcare have increased dramatically. But the operating context for all stakeholders remains influenced by obsolete thinking. There is an urgent need for updating EU guidance to take account of progress and the prospects of greater advances.

## 6. Biomarkers in Action: Clinical Use Cases

It is in oncology that biomarker testing is most evidently proving its worth, in breast, ovarian, prostate, lung, thyroid and colon cancers [11].

In breast cancer, testing of tumour biopsy now makes it possible to identify certain mutations that drive treatment resistance, permitting better risk assessment and aiding the search for targeted therapies [11]. In ovarian cancer, germline mutations in *BRCA1* and *BRCA2* may support identification of potentially at-risk family members through cascade testing, or the offer of germline testing to the direct family members of a positive patient [12,13]. The homologous recombination deficiency (HRD) testing [14] which incorporates new genomic instability assessment algorithms, offers the opportunity to enhance the utility for BRCA testing in ovarian, breast, pancreatic and other cancers [14]. Multi-gene expression profiling can prognose outcomes in early-stage breast cancer and the aggressiveness of prostate cancer, informing treatment decisions, and supporting clinical decision-making inpatients with early stage breast cancer who may require adjuvant chemotherapy, in whom standard of diagnostic care yields an indeterminate prognosis. Finally, in prostate cancer, it can support clinical decision making in patients with indeterminate risk based on other testing data.

In non-small cell lung cancer, tests for mutations identify patients eligible for targeted therapies [15]: Therefore, biomarker testing offers assistance in defining lung cancer options, ranging from optimized image-based screening to the use of autoantibodies or other circulating biomarkers, such as microRNAs and protein panels [11,16]. Molecular testing for colorectal cancer helps adapting treatment to targeted therapies [17]. In thyroid cancer, knowing the oncogenic molecular driver helps to determine the aggressiveness of the tumour and/or to identify the most appropriate systemic or targeted therapy [11]. Across the entire range of cancer—“pan-cancer” as well as tumour agnostics—there is potential in the integration of molecular information, identifying actionable mutations with broad molecular profiling, matching the right targeted therapy to the detected actionable mutation, and evaluating treatment outcome [5,12]. Applications combining testing using next generation sequencing with artificial intelligence and machine learning may ultimately help chart entire clinical pathways [12].

## 7. Tumour-Agnostic Therapies

These potentially game-changing therapies can supersede the “one size fits all” approach in oncology, but they require broad next generation sequencing available across tumour types. Available tumour-agnostic products are already a step towards precision medicine, but regulators, health technology assessment bodies and consumers are cautious over the complexities of demonstrating effectiveness in largely unexplored avenues of evidence generation through still-unfamiliar approaches such as basket trials and innovative methods such as optimized diagnostic algorithms [18]. These therapies are also prompting new demands for wider understanding of the merits of real-word evidence [4].

## 8. Potential Solutions

The list of deficiencies and barriers to exploiting the potential of biomarkers is long, but so too is the list of solutions that are already available—or should be.

Many of the issues relating to funding require only marginal adjustments, via a policy framework to support diagnostics in the EU by 2022, with a ring-fenced budget allowance for biomarker testing development and clinical validation, and the promotion of research investment. But innovative diagnostic technologies lack reimbursement pathways in many countries, and Health Technology Assessment (HTA) mechanisms connected to a reimbursement decision could inspire improvements [19] to develop systems where decision makers, including HTA bodies informed by input from patients, would define evidentiary standards for diagnostics and would commit to pay for products that met them [20].

Much can be achieved by more effective collaboration. The EU should agree by 2023 a business model for public-private cooperation to make optimal biomarker testing available across the EU, based on the business and value case to provide infrastructure to meet testing requirements. This would help provide the evidence base on biomarker testing, and define how it is accessed on a pan-cancer registry across the EU.

Data collection, quality, standards and interoperability need standardisation to allow clinically-relevant biomarkers to be measured and reported. A federated structure of national databases would permit large international multi-centre clinical validation studies of biomarkers (especially early diagnostic and prognostic biomarkers), which require large and long-lasting cohorts.

An EU framework for quality of testing and value of diagnostics information should provide standards for laboratories where samples are collected, stabilised and stored. This would facilitate centralised and standardised registries of diagnosis including sequence and biomarker data, pre-analytical sample metadata, and treatment and outcome data, and could feed into a fast-track approval mechanism for biomarker validation, accompanied by guidance on minimal testing standards and resource allocation [21].

Clarity for clinicians on where and when tests should be performed should be provided, with captured testing data informing service improvements, benchmarking and research. A clinical infrastructure that turned fit-for-purpose real world data into real world evidence would help overcome the deficiencies of existing datasets. And a global or pan-European pan-cancer registry provide insights valuable learning on sharing data across borders [22,23].

Collaboration would engage multiple specialties from drug and diagnostic developers, clinicians, biologists, biostatisticians and digital technology groups, as testing objectives matured from risk assessment to informing treatment decisions. Member state promotion of engagement between payer organizations, biomarker developers and the wider healthcare stakeholder community would ensure that new validated biomarker tests were rapidly made available to patients without the imposition of unrealistic evidentiary burdens.

There are opportunities still to be explored—such as providing simpler testing kits for complex conditions, and notably the development of blood biopsy, or the convergence of complex test offerings with predictive protein-, genetic- and epigenetic-based biomarkers or NGS panels, or the development of predictive potential of prognostic tests (effectively as companion diagnostics) with emerging drugs [11,24,25].

## 9. The Particular Challenges of Tumour Agnostic Therapies (TAx)

The current gaps and possible solutions apply with added intensity to histology-independent tumour agnostic therapies (TAx), which differ from conventional anti-cancer treatments focusing on specific genomic or molecular alterations of cancer cells rather than the tissue of origin. The same drug has potential to be used to treat various unique types of cancer, including very rare tumours, as long as the biomarker targeted by the drug is present [26]. Trials of TAx, especially in rare and ultra-rare populations, face challenges in generating comparative evidence. Nevertheless, low patient numbers for single tumour entities with the specific biomarker mean that statistical proof of effect and estimation of between-tumour heterogeneity are challenging.

The basis of approval for these products is a biomarker present across many tumour types, and biomarkers that measure response, establishing the effects of context, and deciphering mechanisms of treatment resistance across a variety of tumour types, also in the context of basket trials [18,19,20].

Nonetheless, the European Medicines Agency (EMA) and the US Food and Drug Administration (FDA) have considered that a high response rate with a long duration of response in a basket trial can be enough to support a histology independent cancer drug’s efficacy [18,27,28]. But current approaches to value assessment and the diagnostic infrastructure are not adequate, and new approaches have not yet been broadly validated. HTA bodies are increasingly confronted with large uncertainty in the evidence base available to inform coverage and reimbursement decisions [28,29].

## 10. Surveying National Attitudes to Tax

A research regarding the attitudes of clinicians, regulators and academics in several leading countries revealed the challenges related to the use of TAx [27]. It assessed national acceptability of basket trials for HTA, for therapies in general, and specifically in the case of TAx. For TAx, questions related to how far HTA recommendations included efficacy endpoints beyond overall survival, such as quality of life or clearly measurable biomarkers, such as the size of tumour; and whether HTA agencies should accept evidence from uncontrolled, multicentre, open-label, single arm clinical trials [23,29].

The FDA has recently approved two drugs in tissue/site agnostic indications: (a) Pembrolizumab for the treatment of unresectable or metastatic, microsatellite instability high or mismatch-repair-deficient solid tumours, irrespective of tumour site or histology and (b) larotrectinib for solid tumours with a neurotrophic receptor tyrosine kinase gene fusion. A similar application for larotrectinib to the EMA for evaluation was submitted, which received a positive opinion by the Committee for Medicinal Products for Human Use (CHMP). The different pathways for submissions and approval have been deeply detailed by Wilking et al. [28].

It explored healthcare system reliance on post-authorisation evidence generation to facilitate patient access to histology independent cancer drugs, the existence of dedicated HTA pathways for specialised, innovative technologies like TAx, national readiness to grant conditional patient access until a final recommendation is made on the basis of further evidence is available (such as after the finalisation of post authorisation studies, or through the UK Cancer Drugs Fund, and opportunities for conditional reimbursement arrangements, in general, and specifically for Tax [19,29].

On testing, the survey inquired into whether routine availability of diagnostic tests—such as broad panel NGS—is considered a prerequisite for TAx, and whether mechanisms for reimbursement of diagnostic tests (such as broad panel NGS or WGS) should be separate from HTA/reimbursement assessments of TAx.

Frequently mentioned barriers to assessing TAx included the lack of comparative effectiveness, poorly characterized prognostic value of the genomic alteration defining the tumour-agnostic approach, and limited knowledge on natural history of identified patients’ populations [25,26,29]. Lack of clarity in the diagnosis pathway, the use of surrogate endpoints (without evidence on the drug’s efficacy on progression-free survival and overall survival), inappropriate design, low prevalence or low number of patients enrolled in studies, often with heterogeneity of previous treatment, were also highlighted. The innate complexity of the healthcare sector and lack of appropriate regulation were seen as further obstacles [25].

Our survey detected some willingness to create alignment between regulators and HTA bodies in terms of evidence sources, endpoint requirements and acceptability of relevant post-authorisation data collection models, but there was wide recognition of a lack of consensus over priorities. There appears at present little common ground on the potentially influential factors to improve the situation—across a range that runs from government support to adequate infrastructure, via involvement of key stakeholder groups, increased awareness and understanding of TAx, and adaptive HTA processes. Crucially, there is currently only little movement to create alignment between regulators and HTA bodies in terms of evidence sources, endpoint requirements and acceptability of relevant post authorisation data collection models. Furthermore, readiness in principle to evolve managed entry agreements beyond discounts is not widely matched by corresponding action. Similarly, appropriate diagnostic and data infrastructure is not available everywhere, and is not a priority everywhere.

The experts from the, Canada, France, Germany, Italy, Spain and UK have contributed to an OHE report of serious gaps in awareness levels among HTA agencies and reluctance to explore innovative payment mechanisms that go beyond discounts. They commented in particular on French, German, Canadian and Italian scepticism over evidentiary support for claims, divergent responses to basket trials in Canada and Italy, and resistance to TAx-friendly regulation and reimbursement in the UK, Spain and South Korea, The general absence of widespread and economically acceptable testing constrains progress in TAx, it was concluded. Although *BRCA1/2* and HRD testing represent a paradigm of personalized treatment in ovarian cancer, the immediate translation to other cancer still challenging due to some financial and technical issues, the latter particularly related to the failure of test when applied to low-quality tissues, as it happened in one third of FFPE sample analysed in the the PROfoud study at ESMO 2019 [30].

The lack of comparative data was however perceived as an understandable—and at present inevitable—challenge, given the nature of the new entities under study from a molecular point of view. The immediate issue, therefore, is to identify the level of uncertainty the different agencies are ready to tolerate, since unflinching insistence on historical methods for assessing drugs precludes any chance of progress.

Early dialogue is seen as key to mitigating uncertainty, through initial discussion with relevant bodies at European level to evolve some shared views that could be translated to the national level and have some influence on discussions on price and reimbursement. Discussion of uncertainty from a clinical point of view has also led to—for instance—a framework developed by ESMO to rank genomic alteration, and work of this type could help advance understanding of the genomic alteration with respect to different cancer types. Giving specific weight to alterations could help in prioritising treatment options [20,27].

## 11. Real-World Evidence (RWE)

The discussions of data adequacy are already turning towards the use of real-world evidence, and highlighting the need for better access so as to confirm where therapies are bringing value to patients. Fragmentation and low prevalence of appropriately digitalised systems currently inhibit collection of data, with many hospitals and centres still focused on financial accounting and working on paper. Policy decisions are needed to initiate improvements, and funding is a further challenge [23,24,25,26,27,28,29,30,31].

There are signs of some longer-term thinking and even some policy shifts to improve the situation. The Organisation for Economic Co-operation and Development (OECD) health ministers have urged health data governance frameworks to ease use of personal health data for health-related public interest purposes with common data elements and formats [7]. Challenges in developing RWE include informed consent, and accuracy of data collected in a “real world” setting. But post-marketing, it can identify adverse effects in a larger population and in population sub-groups, and contribute to a broader long-term follow up, on condition of adequate statistical methods to extract, analyse and interpret real-world data (RWD). Opportunities for patients are lost when science is progressing faster than the “system”, impeding the development and best use of novel treatment options. Instead of considering that analysing personal health data is a risk to individuals, it should be accepted that the reverse is the case: Not analysing personal health data is a risk to individuals [23,25].

There are a growing number of initiatives to create a federated, interoperable, cross border data infrastructure that could advance RWE. Examples include Gaia-x [32], led by figures form politics, business and science in France and Germany, or Medical Informatics in Germany a multi-stakeholder consortium created to close the gap between research and healthcare [33]. MedMij in the Netherlands is promoting digital exchange of personal health data [34], and the French government’s HealthdataHub cross-references health databases for researchers [35,36,37,38]. A recent change in US legislation also combats fragmentation by limiting the scope for holders of data to actively prevent sharing [39]. A change to the US CMS final rule requires Medicare and Medicaid participating hospitals to share electronic notifications with other providers.

## 12. Tentative Conclusions

Ultimately, successful development and deployment of biomarker testing depends on a policy framework in which countries would find it easier to reach consistent decisions and to provide clearer funding arrangements, thus boosting access and continued development. The EU should take the lead in developing or promoting clear and updated guidance for regulators and payers/customers, for public and private laboratories, for clinicians and healthcare providers, on the active development and use of biomarker testing.

There are some signals that could justify guarded hopes of an improved environment. Recent demonstrations of wide support for EU initiatives such as its Beating Cancer Plan or its Cancer Mission, as well as numerous declarations made by the EU institutions both before and during the coronavirus crisis, suggest a growing recognition of the need to innovate—at the level of both policymakers and of the health community. The renewed attention to disparities in cancer care and access across Europe is also driving new assessments of obstacles and new pursuits of solutions, and promoting greater networking and collaboration among cancer institutions.

But nothing will happen by accident. Constructive change to the health care context could ensure better use of the potential offered by new technologies in testing, in diagnosis and in treatment of cancer, through development and use of biomarkers and the advanced treatments—such as personalised medicine and TAx—that they enable. But this will result only from vigorous debate among all stakeholders, and agreement on recommendations of a technical and political nature that will result in a better deal for patients and a more sustainable approach to healthcare.

## 13. Recommendations

### 13.1. Recommendations to the EU

Update the recommendation on early detection strategies allowing for risk stratification through molecular diagnostics/biomarker testing.Provide guidance to member states on minimal testing standards (also in light of IVDR), and on creation of a systematic reference framework for duly-accredited clinical laboratories.Establish a framework promoting the value of diagnostics information and accredited methodologies for cancer detection and treatment.Create a fast track approval mechanism for biomarker validation.Consider a 12 month postponement for application of the IVDR.

### 13.2. Recommendations to EMA

Early engagement on regulatory qualification of novel biomarkers, and review the biomarker clinical and analytical validation process.Develop multi-stakeholder scientific advice on the use of companion diagnostics.Align the post-launch TAx evidence generation commitments with HTAs and pharma companies.Promote consensus on design and set up of confirmatory basket trials.

### 13.3. Recommendations to Member States

Ensure that new validated biomarker tests are rapidly made available to patients.Incentivize the development and uptake of biomarkers of limited interest to commercial companies (early detection biomarkers & risk biomarkers).Synchronise CDx approval processes with drug approval.Promote alignment between national regulators and payers/customers on standardized outcome measures, systematic data collection, and data standards and sharing, and integrate registries for TAx with rare mutations.Cooperate on a federated structure of national databases for robust EU-wide data-sets with uniform criteria and formats for research and real-world data, and standardized registries of genomics and outcome data.

### 13.4. Recommendations to All Stakeholders

Cooperate on information and education, including literacy for the public and for professionals, with a short-term focus on influencers, notably payers.Cooperate on pan-cancer studies, with more, and more targeted, screening, exploiting the potential of stratification and of genomics, AI, biomarker testing.

### 13.5. Recommendations on Funding

The EU to agree by 2023 a business model for public-private cooperation for optimal biomarker testing available across the EU.European health authorities to put in place a policy framework to support diagnostics in the EU by 2022, with a ring-fenced budget allowance for biomarker testing development (clinical validation).Member states to allocate resources specifically for discovery and validation of biomarkers, and promote engagement between payer organizations, biomarker developers and the wider healthcare stakeholder community. (vertical integration and horizontal integration).Adaptive reimbursement pathways to be linked to conditional reimbursement based on evidence development for TAx, with uncertainties mitigated through managed entry agreements.Research to be promoted on biomarkers discovery and early testing.

## Data Availability

Please visit the Website: www.euapm.eu.

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
