# Peer review of "Bringing Onco-Innovation to Europe’s Healthcare Systems: The Potential of Biomarker Testing, Real World Evidence, Tumour Agnostic Therapies to Empower Personalised Medicine"

_cancers, 2021, doi:10.3390/cancers13030583_

Round 1

Reviewer 1 Report

The submitted manuscript sounds more like an Opinion of a very large group of specialists (48) with a set of specific Recommendations for health organizations in a certain part of the world. It can be published as a conference paper or detailed guidelines for professional organization/society. In its current form, the manuscript doesn't meet the criteria for either Article or Review categories as specified by the Cancers journal:

  • Articles: Original research manuscripts. The journal considers all original research manuscripts, provided that the work reports scientifically sound experiments and provides a substantial amount of new information, e.g., research articles using only one cell line for the experiments will not be considered for publication (experiments need to be repeated on 1-2 more cell lines); authors should consider in vivo studies using orthotopic or transgenic models to validate gene function; for all Western blot figures, densitometry readings/intensity ratio of each band should be included; the whole Western blot showing all bands and molecular weight markers should be included in the Supplementary Materials; gene silencing experiments should use at least two gene-specific siRNAs, etc.

    Full experimental details must be provided so that the results can be reproduced. Cancers requires that authors publish all experimental controls and make full datasets available where possible (see the guidelines on Supplementary Materials and references to unpublished data).

    Articles should have a main text of around 3000 words at minimum and should have more than 30 references. Cancers has no restrictions on the maximum length of research manuscripts, provided that the text is concise and comprehensive.

  • Reviews: These provide concise and precise updates on the latest progress made in a given area of research. Systematic reviews should follow the PRISMA guidelines. Review articles should be comprehensive and submitted by authors who are in the field. The main text of review papers should be around 4000 words at minimum and include at least two figures or tables.

Author Response

The submitted manuscript sounds more like an Opinion of a very large group of specialists (48) with a set of specific Recommendations for health organizations in a certain part of the world. It can be published as a conference paper or detailed guidelines for professional organization/society. In its current form, the manuscript doesn't meet the criteria for either Article or Review categories as specified by the Cancers journal.

We thank the reviewer for his comments, but we partially agree with the view that this paper looks   like an Opinion paper. Before submitting it, we held some advisory boards in order to share our country or regional experiences and verify if there is an opportunity to harmonize the path for cancer patients in the European countries. We have verified that, sometimes, in the same countries, depending on the regions and regional rules, the patients’ paths are different. We cannot accept this view and we need to publish the results of this paper in order to push the European Parliament to increase the budget availability in order the cover in order to cover all the highest possible number of patients. Therefore, I believe that this paper can achieve the criteria of an Article, rather than a review manuscript. I believe that the Editor of the Special issue should decide on the attribution of the most suitable type of article.

Number of co-authors has been reduced.

English: text has been revised, also considering that the first author (the main writer) is a native English speaker

Reviewer 2 Report

The authors present an argument for updating the EU policies and frameworks surrounding access to appropriate biomarker testing in cancer and other diseases.

The document is well written, if a little repetitious. The document is lacking in context surrounding how and why this group specifically came together to write this document. Is it a summary of a working group meeting? Conference? Did the authors action the survey mentioned? This needs to be spelt out for clarity and context. I think it would be of value to describe a system that functions in a a close to ideal way - be it MSK in their integration of research within the clinical framework, or some other countries with centralised testing etc etc. It would be nice to see what you are aiming for - even as a picture.

Referencing should be increased throughout, as should definition of abbreviations.

Specific comments:

The authors present an argument for updating the EU policies and frameworks surrounding access to appropriate biomarker testing in cancer and other diseases.

The document is well written, if a little repetitious. The document is lacking in context surrounding how and why this group specifically came together to write this document. Is it a summary of a working group meeting? Conference? Did the authors action the survey mentioned? This needs to be spelt out for clarity and context. I think it would be of value to describe a system that functions in a a close to ideal way - be it MSK in their integration of research within the clinical framework, or some other countries with centralised testing etc etc. It would be nice to see what you are aiming for - even as a picture.

Referencing should be increased throughout, as should definition of abbreviations.

Specific comments:

Is reference 1 the most appropriate? I wouldn't have thought so.

Line 85- this would be backed by clarity over reimbursement – I do not understand this comment.

Line 119 – pan cancer studies – do you mean research? Or Clinical programs?

Line 123 – prognose for = prognosticate

Section Tumour Agnostic Therapies –It would be useful to provide global context here – what are other countries doing that the EU should be aspiring to

Line 166– define HRD testing, and reference the concept and its implementation

Paragraph starting on line 168 needs referencing

Define HTA? Line 205

The tumour agnostic sections need to be reviewed for repetition.

Line 267 – why not reference some of the Memorial Sloane Kettering publications on basket trials

Line 281 – reference the survey – more info – who conducted it? Were there any quantifiable outcomes. This section is very conversational and it is not clear what the key messages are. Synthesis would be helpful - are most EU countries in the same boat?

Line 329 – this response is most definitely not clear! Reconsider what exactly you hoped to exemplify with it and perhaps paraphrase?

 ‘Payers’ is not standard language – do you mean consumers?

I have a few comments:

  1. The group needs to define how and why they have come together to write the proposal - helps to give context and allows readers to make their own assessments on bias/conflict
  2. It still remains unclear who executed the survey and under what conditions etc
  3. Line 343 - Real World Evidence - need to give examples of how the process should work - maybe refs to US world leaders in tumour agnostic trials etc- MSKCC, DFCI, others
  4. payers is correct - perhaps consumers?
  5. some revision of repetition is still necessary

Author Response

The authors present an argument for updating the EU policies and frameworks surrounding access to appropriate biomarker testing in cancer and other diseases.

The document is well written, if a little repetitious. The document is lacking in context surrounding how and why this group specifically came together to write this document. Is it a summary of a working group meeting? Conference? Did the authors action the survey mentioned? This needs to be spelt out for clarity and context. I think it would be of value to describe a system that functions in  a close to ideal way - be it MSK in their integration of research within the clinical framework, or some other countries with centralised testing etc etc. It would be nice to see what you are aiming for - even as a picture.

Response: we focused more on the context from which this work was born submitting it, we held some advisory boards in order to share our country or regional experiences and verify if there is an opportunity to harmonize the path for cancer patients in the European countries. We have verified that, sometimes, in the same countries, depending on the regions and regional rules, the patients’ paths are different. We cannot accept this view and we need to publish the results of this paper in order to push the European Parliament to increase the budget availability in order the cover in order to cover all the highest possible number of patients. No conflict of interest was present. Therefore, I believe that this paper can achieve the criteria of an Article, rather than a review manuscript. I believe that the Editor of the Special issue should decide on the attribution of the most suitable type of article.

Referencing should be increased throughout, as should definition of abbreviations.

 We tried to eliminate repetition within the text.

Specific comments:

Is reference 1 the most appropriate? I wouldn't have thought so. We have update the Reference 1

Line 85- this would be backed by clarity over reimbursement – I do not understand this comment. Sentence has been sligjtly modified

Line 119 – pan cancer studies – do you mean research? Or Clinical programs? We have changed with research

Line 123 – prognose for = prognosticate We have changed as suggested

Section Tumour Agnostic Therapies –It would be useful to provide global context here – what are other countries doing that the EU should be aspiring to- We have included a new reference in this regard.

Line 166– define HRD testing, and reference the concept and its implementation. A new reference has been added)

Paragraph starting on line 168 needs referencing [references 6 and 7 covers the two paragraphs]

Define HTA? Line 205 Done

The tumour agnostic sections need to be reviewed for repetition.  AS requested, the text has been shortened

Line 267 – why not reference some of the Memorial Sloane Kettering publications on basket trials. A specific reference has been added.

Now reference

Line 281 – reference the survey – more info – who conducted it? Were there any quantifiable outcomes. This section is very conversational and it is not clear what the key messages are. Synthesis would be helpful - are most EU countries in the same boat? Reference has been included

Line 329 – this response is most definitely not clear! Reconsider what exactly you hoped to exemplify with it and perhaps paraphrase? Sentence has been rephrased

 ‘Payers’ is not standard language – do you mean consumers?

Payers can be both customers and puclic or private institutions. In this meaning we have now reported payer/customers

 I have a few comments: We have already elucidated through the above responses and text revisions the questions reported below.

Reviewer 3 Report

The revised manuscript is appropriate for publication.

Author Response

Thanks a lot for your positive evaluation.

Reviewer 4 Report

This extensive review by Horgan et al. summarizes benefits of tumor agnostic testing and discusses BRCA1/2 testing as an example. The group involves numerous scientists from different countries. Overall, the review is well written.

I have several minor points that needs to be addressed:

1-Please describe how the literature research was conducted.

2-Since this a pretty big group that has scientists from different EU countries, including special examples in terms of state of the agnostic testing from these countries would be very informative. Examples were given from UK (not technically in EU anymore), Canada, Germany, Spain, Italy and South Korea. For example a table would be very helpful. 

3-A comparison with other developed countries (USA, Australia etc.) helps reader to understand where EU is situated currently better.

Author Response

This extensive review by Horgan et al. summarizes benefits of tumor agnostic testing and discusses BRCA1/2 testing as an example. The group involves numerous scientists from different countries. Overall, the review is well written.

I have several minor points that needs to be addressed:

1-Please describe how the literature research was conducted.

Literature was selected with specific queries including agnostic therapies, oncogenetic, health technology assessment, reimbursement, harmonization, cancer genomic testing, European countries, above all focusing on those papers mainly related to the application of model of HTA and Personalized medicine in cancer setting. Obviously, since literature is full of papers, we selected mainly those with matched with the aim of our multidisciplinary working group. All references have been reviewed and updated as suggested.  

2-Since this a pretty big group that has scientists from different EU countries, including special examples in terms of state of the agnostic testing from these countries would be very informative. Examples were given from UK (not technically in EU anymore), Canada, Germany, Spain, Italy and South Korea. For example a table would be very helpful. 

Regarding this aspect, we refer to the publication by Wilking N, et al. ESMO Open 2019;4:e000550. doi:10.1136/esmoopen-2019-000550, now reported in the revise version of paper.

3-A comparison with other developed countries (USA, Australia etc.) helps reader to understand where EU is situated currently better.

We have added a sentence regarding the use of innovative drugs in US and Europe.

Round 2

Reviewer 2 Report

Line 72 ‘that’ emerged

Line 90 ‘that’ emerged

Line 213 – I see customers has been added but ‘payers’ is not used in this context in English. Why not say consumers?

Line 300 this sentence is still confusing.

TAx – ensure this is used throughout, not Tax